

# The Oyster River Protocol: a multi-assembler and kmer approach for de novo transcriptome assembly

Matthew D. MacManes

Department of Molecular, Cellular and Biomedical Sciences, University of New Hampshire, Durham, NH, USA

## ABSTRACT

Characterizing transcriptomes in non-model organisms has resulted in a massive increase in our understanding of biological phenomena. This boon, largely made possible via high-throughput sequencing, means that studies of functional, evolutionary, and population genomics are now being done by hundreds or even thousands of labs around the world. For many, these studies begin with a de novo transcriptome assembly, which is a technically complicated process involving several discrete steps. The Oyster River Protocol (ORP), described here, implements a standardized and benchmarked set of bioinformatic processes, resulting in an assembly with enhanced qualities over other standard assembly methods. Specifically, ORP produced assemblies have higher Detonate and TransRate scores and mapping rates, which is largely a product of the fact that it leverages a multi-assembler and kmer assembly process, thereby bypassing the shortcomings of any one approach. These improvements are important, as previously unassembled transcripts are included in ORP assemblies, resulting in a significant enhancement of the power of downstream analysis. Further, as part of this study, I show that assembly quality is unrelated with the number of reads generated, above 30 million reads. Code Availability: The version controlled open-source code is available at https://github.com/macmanes-lab/Oyster_River_Protocol. Instructions for software installation and use, and other details are available at http://oyster-river-protocol.rtfd.org/.

## INTRODUCTION

For all biology, modern sequencing technologies have provided for an unprecedented opportunity to gain a deep understanding of genome level processes that underlie a very wide array of natural phenomena, from intracellular metabolic processes to global patterns of population variability. Transcriptome sequencing has been influential (*Mortazavi et al., 2008*; *Wang, Gerstein & Snyder, 2009*), particularly in functional genomics (*Lappalainen et al., 2013*; *Cahoy et al., 2008*), and has resulted in discoveries not possible even just a few years ago. This in large part is due to the scale at which these studies may be conducted (*Li et al., 2017*; *Tan et al., 2017*). Unlike studies of adaptation based on one or a small number of candidate genes (*Fitzpatrick et al., 2005*;

Corresponding author
Matthew D. MacManes,
macmanes@gmail.com

*Panhuis, 2006*), modern studies may assay the entire suite of expressed transcripts—the transcriptome—simultaneously. In addition to issues of scale, as a direct result of enhanced dynamic range, newer sequencing studies have increased ability to simultaneously reconstruct and quantitate lowly- and highly-expressed transcripts (*Wolf, 2013*; *Vijay et al., 2013*). Lastly, improved methods for the detection of differences in gene expression (*Robinson, McCarthy & Smyth, 2010*; *Love, Huber & Anders, 2014*) across experimental treatments have resulted in increased resolution for studies aimed at understanding changes in gene expression.

As a direct result of their widespread popularity, a diverse tool set for the assembly of transcriptome exists, with each potentially reconstructing transcripts others fail to reconstruct. Amongst the earliest of specialized de novo transcriptome assemblers were the packages `Trans-ABySS` (*Robertson et al., 2010*), `Oases` (*Schulz et al., 2012*), and `SOAPdenovoTrans` (*Xie et al., 2014*), which were fundamentally based on the popular de Bruijn graph-based genome assemblers `ABySS` (*Simpson et al., 2009*), `Velvet` (*Zerbino & Birney, 2008*), and `SOAP` (*Li et al., 2008*), respectively. These early efforts gave rise to a series of more specialized de novo transcriptome assemblers, namely `Trinity` (*Haas et al., 2013*), and `IDBA-Tran` (*Peng et al., 2013*). While the de Bruijn graph approach remains powerful, newly developed software explores novel parts of the algorithmic landscape, offering substantial benefits, assuming novel methods reconstruct different fractions of the transcriptome. `BinPacker` (*Liu et al., 2016*), for instance, abandons the de Bruijn graph approach to model the assembly problem after the classical bin packing problem, while `Shannon` (*Kannan et al., 2016*) uses information theory, rather than a set of software engineer-decided heuristics. These newer assemblers, by implementing fundamentally different assembly algorithms, may reconstruct fractions of the transcriptome that other assemblers fail to accurately assemble.

In addition to the variety of tools available for the de novo assembly of transcripts, several tools are available for pre-processing of reads via read trimming (e.g., `Skewer`; *Jiang et al., 2014*, `Trimmomatic`; *Bolger, Lohse & Usadel, 2014*, `Cutadapt`; *Martin, 2011*), read normalization (`khmer`; *Pell et al., 2012*), and read error correction (`SEECER`; *Le et al., 2013*, `RCorrector`; *Song & Florea, 2015*, `Reptile`; *Yang, Dorman & Aluru, 2010*). Similarly, benchmarking tools that evaluate the quality of assembled transcriptomes including `TransRate` (*Smith-Unna et al., 2016*), `BUSCO` (Benchmarking Universal Single-Copy Orthologs; *Simão et al., 2015*), and `Detonate` (*Li et al., 2014*) have been developed. Despite the development of these evaluative tools, this manuscript describes the first systematic effort coupling them with the development of a de novo transcriptome assembly pipeline.

The ease with which these tools may be used to produce and characterize transcriptome assemblies belies the true complexity underlying the overall process (*Ungaro et al., 2017*; *Wang & Gribskov, 2017*; *Moreton, Izquierdo & Emes, 2015*; *Yang & Smith, 2013*). Indeed, the subtle (and not so subtle) methodological challenges associated with transcriptome reconstruction may result in highly variable assembly quality. In particular, while most tools run using default settings, these defaults may be sensible only for one specific (often unspecified) use case or data type. Because parameter optimization is both

dataset-dependent and factorial in nature, an exhaustive optimization particularly of entire pipelines, is never possible. Given this, the production of a de novo transcriptome assembly requires a large investment in time and resources, with each step requiring careful consideration. Here, I propose an evidence-based protocol for assembly that results in the production of high quality transcriptome assemblies, across a variety of commonplace experimental conditions or taxonomic groups.

[1] Named the Oyster River Protocol because the ideas, and some of the code, was developed while overlooking the Oyster River, located in Durham, New Hampshire. NB, the naming assembly of protocols after bodies of water was, to the best of my knowledge, first done by C. Titus Brown (The Eel Pond Protocol: http://khmer-protocols.readthedocs.io/en/latest/mrnaseq/index.html), and may have subconsciously influenced me in naming this protocol.

This manuscript describes the development of The Oyster River Protocol (ORP)[1] for transcriptome assembly. It explicitly considers and attempts to address many of the shortcomings described in *Vijay et al. (2013)*, by leveraging a multi-kmer and multi-assembler strategy. This innovation is critical, as all assembly solutions treat the sequence read data in ways that bias transcript recovery. Specifically, with the development of assembly software comes the use of a set of heuristics that are necessary given the scope of the assembly problem itself. Given each software development team carries with it a unique set of ideas related to these heuristics while implementing various assembly algorithms, individual assemblers exhibit unique assembly behavior. By leveraging a multi-assembler approach, the strengths of one assembler may complement the weaknesses of another. In addition to biases related to assembly heuristics, it is well known that assembly kmer-length has important effects on transcript reconstruction, with shorter kmers more efficiently reconstructing lower-abundance transcripts relative to more highly abundant transcripts. Given this, assembling with multiple different kmer lengths, then merging the resultant assemblies may effectively reduce this type of bias. Recognizing these issue, I hypothesize that an assembly that results from the combination of multiple different assemblers and lengths of assembly-kmers will be better than each individual assembly, across a variety of metrics.

In addition to developing an enhanced pipeline, the work suggests an exhaustive way of characterizing assemblies while making available a set of fully-benchmarked reference assemblies that may be used by other researchers in developing new assembly algorithms and pipelines. Although many other researchers have published comparisons of assembly methods, up until now these have been limited to single datasets assembled a few different ways (*Marchant et al., 2016*; *Finseth & Harrison, 2014*), thereby failing to provide more general insights.

## METHODS

### Datasets

In an effort at benchmarking the assembly and merging protocols, I downloaded a set of publicly available RNAseq datasets (Table 1) that had been produced on the Illumina sequencing platform. These datasets were chosen to represent a variety of taxonomic groups, so as to demonstrate the broad utility of the developed methods. Because datasets were selected randomly with respect to sequencing center and read number, they are likely to represent the typical quality of Illumina data circa 2014–2017.

### Software

The ORP can be installed on the Linux platform, and does not require superuser privileges, assuming `Linuxbrew` (*Jackman & Birol, 2016*) is installed. The software is implemented

**Table 1 Lists the datasets used in this study.**

| Type | Accession | Species | Number of reads (M) | Read length (bp) |
|------|-----------|---------|---------------------|------------------|
| Animalia | ERR489297 | *Anopheles gambiae* | 206 | 100 |
| Animalia | DRR030368 | *Echinococcus multilocularis* | 73 | 100 |
| Animalia | ERR1016675 | *Heterorhabditis indica* | 51 | 100 |
| Animalia | SRR2086412 | *Mus musculus* | 54 | 100 |
| Animalia | DRR036858 | *Mus musculus* | 114 | 100 |
| Animalia | DRR046632 | *Oncorhynchus mykiss* | 82 | 76 |
| Animalia | SRR1789336 | *Oryctolagus cuniculus* | 31 | 100 |
| Animalia | SRR2016923 | *Phyllodoce medipapillata* | 86 | 100 |
| Animalia | ERR1674585 | *Schistosoma mansoni* | 39 | 100 |
| Plant | DRR082659 | *Aeginetia indica* | 69 | 90 |
| Plant | DRR053698 | *Cephalotus follicularis* | 126 | 90 |
| Plant | DRR069093 | *Hevea brasiliensis* | 103 | 100 |
| Plant | SRR3499127 | *Nicotiana tabacum* | 30 | 150 |
| Plant | DRR031870 | *Vigna angularis* | 60 | 100 |
| Protozoa | ERR058009 | *Entamoeba histolytica* | 68 | 100 |

**Note:**
All datasets are publicly available for download by accession number at the European Nucleotide Archive or NCBI Short Read Archive.

as a stand-alone makefile which coordinates all steps described below. All scripts are available at https://github.com/macmanes-lab/Oyster_River_Protocol, and run on the Linux platform. The software is version controlled and openly-licensed to promote sharing and reuse. A guide for users is available at http://oyster-river-protocol.rtfd.io.

## Pre-assembly procedures

For all assemblies performed, Illumina sequencing adapters were removed from both ends of the sequencing reads, as were nucleotides with quality Phred ≤2, using the program `Trimmomatic` version 0.36 (*Bolger, Lohse & Usadel, 2014*), following the recommendations from *MacManes (2014)*. After trimming, reads were error corrected using the software `RCorrector` version 1.0.2 (*Song & Florea, 2015*), following recommendations from *MacManes & Eisen (2013)*. The code for running this step of the ORPs is available at https://github.com/macmanes-lab/Oyster_River_Protocol/blob/master/oyster.mk#L145. The trimmed and error corrected reads were then subjected to de novo assembly.

## Assembly

I assembled each trimmed and error corrected dataset using three different de novo transcriptome assemblers and three different kmer lengths, producing four unique assemblies. First, I assembled the reads using `Trinity` release 2.4.0 (*Haas et al., 2013*), and default settings ($k = 25$), without read normalization. The decision to forgo normalization is based on previous work (*MacManes, 2015*) showing slightly worse

performance of normalized datasets. Next, the `SPAdes` RNAseq assembler (version 3.10) (*Chikhi & Medvedev, 2014*) was used, in two distinct runs, using kmer sizes 55 and 75. Lastly, reads were assembled using the assembler `Shannon` version 0.0.2 (*Kannan et al., 2016*), using a kmer length of 75. These assemblers were chosen based on the fact that they (1) use an open-science development model, whereby end-users may contribute code, (2) are all actively maintained and are undergoing continuous development, and (3) occupy different parts of the algorithmic landscape.

This assembly process resulted in the production of four distinct assemblies. The code for running this step of the ORPs is available at https://github.com/macmanes-lab/Oyster_River_Protocol/blob/master/oyster.mk#L148.

## Assembly merging via OrthoFuse

To merge the four assemblies produced as part of the ORP, I developed new software that effectively merges transcriptome assemblies. Described in brief, `OrthoFuse` begins by concatenating all assemblies together, then forms groups of transcripts by running a version of `OrthoFinder` (*Emms & Kelly, 2015*) packaged with the ORP, modified to accept nucleotide sequences from the merged assembly. These groupings represent groups of homologous transcripts. While isoform reconstruction using short-read data is notoriously poor, by increasing the inflation parameter by default to $I = 4$, it attempts to prevent the collapsing of transcript isoforms into single groups. After `Orthofinder` has completed, a modified version of `TransRate` version 1.0.3 (*Smith-Unna et al., 2016*) which is packaged with the ORP, is run on the merged assembly, after which the best (= highest contig score) transcript is selected from each group and placed in a new assembly file to represent the entire group. The resultant file, which contains the highest scoring contig for each orthogroup, may be used for all downstream analyses. `OrthoFuse` is run automatically as part of the ORP, and additionally is available as a stand alone script, https://github.com/macmanes-lab/Oyster_River_Protocol/blob/master/orthofuser.mk.

## Assembly evaluation

All assemblies were evaluated using `ORP-TransRate`, `Detonate` version 1.11 (*Li et al., 2014*), `shmlast` version 1.2 (*Scott, 2017*), and `BUSCO` version 3.0.2 (*Simão et al., 2015*). `TransRate` evaluates transcriptome assembly contiguity by producing a score based on length-based and mapping metrics, while `Detonate` conducts an orthogonal analysis, producing a score that is maximized by an assembly that is representative of input sequence read data. `BUSCO` evaluates assembly content by searching the assemblies for conserved single copy orthologs found in all Eukaryotes. I report default `BUSCO` metrics as described in *Simão et al. (2015)*. Specifically, "complete orthologs," are defined as query transcripts that are within two standard deviations of the length of the `BUSCO` group mean, while contigs falling short of this metric are listed as "fragmented." `Shmlast` implements the conditional reciprocal best hits test (*Aubry et al., 2014*), conducted in this case against the Swiss-Prot protein database (downloaded October, 2017) using an *e*-value of 1$E$-10.

In addition to the generation of metrics to evaluation the quality of transcriptome assemblies, I generated a distance matrix of assemblies for each dataset using the `sourmash` package (*Titus Brown & Irber, 2016*), in an attempt at characterizing the algorithmic landscape of assemblers. Specifically, each assembly was characterized using the `compute` function using 5,000 independent sketches. The distance between assemblies was calculated using the compare function and a kmer length of 51. These distance matrices were visualized using the `isoMDS` function of the MASS package (https://CRAN.R-project.org/package=MASS).

### Statistics

All statistical analyses were conducted in R version 3.4.0 (*R Core Development Team, 2011*). Violin plots were constructed using the beanplot (*Kampstra, 2008*) and the beeswarm R packages (https://CRAN.R-project.org/package=beeswarm). Expression distributions were plotted using the ggridges package (https://CRAN.R-project.org/package=ggridges).

## RESULTS AND DISCUSSION

A total of 15 RNAseq datasets, ranging in size from (30–206 M paired end reads) were assembled using the ORP and with `Trinity`. Each assembly was evaluated using the software BUSCO, `shmlast`, `Detonate`, and `TransRate`. From these, several metrics were chosen to represent the quality of the produced assemblies. Of note, all the assemblies produced as part of this work are available at DOI 10.5281/zenodo.1320141. A file containing the evaluative metrics is available at https://github.com/macmanes-lab/Oyster_River_Protocol/blob/master/manuscript/orp.csv, while the distance matrices are available within the folder https://github.com/macmanes-lab/Oyster_River_Protocol/blob/master/manuscript/. R code used to conduct analyses and make figures is found at https://github.com/macmanes-lab/Oyster_River_Protocol/blob/master/manuscript/R-analysis.Rmd.

### Assembled transcriptomes

The `Trinity` assembly of trimmed and error corrected reads generally completed on a standard Linux server using 24 cores, in less than 24 h. RAM requirement is estimated to be close to 0.5 Gb per million paired-end reads. The assemblies on average contained 176 k transcripts (range 19–643 k) and 97 Mb (range 14 MB–198 Mb). Other quality metrics will be discussed below, specifically in relation to the ORP produced assemblies.

Oyster River Protocol assemblies generally completed on a standard Linux server using 24 cores in 3 days. Typically `Trinity` was the longest running assembler, with the individual `SPAdes` assemblies being the shortest. RAM requirement is estimated to be 1.5–2 Gb per million paired-end reads, with `SPAdes` requiring the most. The assemblies on average contained 153 k transcripts (range 23–625 k) and 64 Mb (range 8 MB–181 Mb).

MinHash sketch signatures (*Ondov et al., 2016*) of each assemblies of a given dataset were calculated using `sourmash` (*Titus Brown & Irber, 2016*), and a MDS plot was

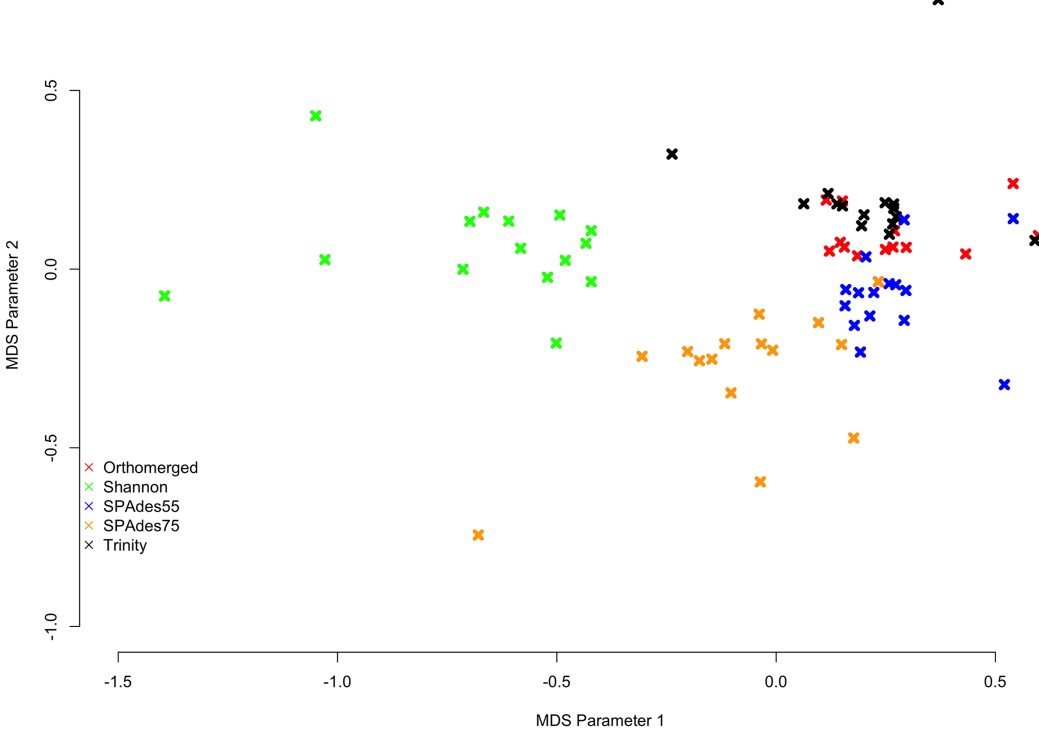

**Figure 1 MDS plot describing the similarity within and between assemblers.** Colored *x*'s mark individual assemblies, with red marks corresponding to the ORP assemblies, green marks corresponding to the Shannon assemblies, blue marks corresponding to the SPAdes55 assemblies, orange marks corresponding to the SPAdes75 assemblies, and the black marks corresponding to the Trinity assemblies. In general assemblies produced by a given assembler tend to cluster together.

generated (Fig. 1) from their distances. Interestingly, each assembler tends to produce a specific signature which is relatively consistent between the fifteen datasets. `Shannon` differentiates itself from the other assemblers on the first (*x*) MDS axis, while the other assemblers (`SPAdes` and `Trinity`) are separated on the second (*y*) MDS axis.

### Assembly structure

The structural integrity of each assembly was evaluated using the `TransRate` and `Detonate` software packages. As many downstream applications depend critically on accurate read mapping, assembly quality is correlated with increased mapping rates. The split violin plot presented in Fig. 2A visually represents the mapping rates of each assembly, with lines connecting the mapping rates of datasets assembled with `Trinity` and with the ORP, respectively. The average mapping rate of the `Trinity` assembled datasets was 87% (sd = 8%), while the average mapping rates of the ORP assembled datasets was 93% (sd = 4%). This test is statistically significant (one-sided Wilcoxon rank sum test, $p = 2E\text{-}2$). Mapping rates of the other assemblies are less than that of the ORP assembly, but in most cases, greater than that of the Trinity assembly. This aspect of assembly quality is critical. Specifically mapping rates measure how representative the assembly is of the reads. If I assume that the vast majority of generated reads come

**Mapping Rate**

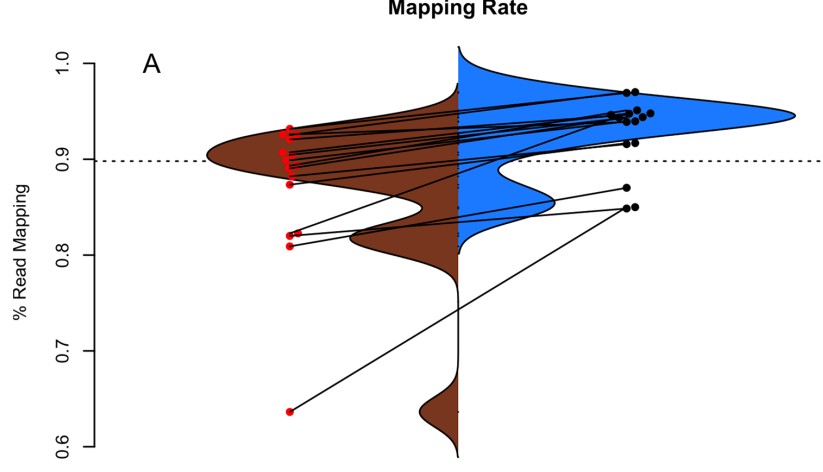

**Optimal Assembly Score**

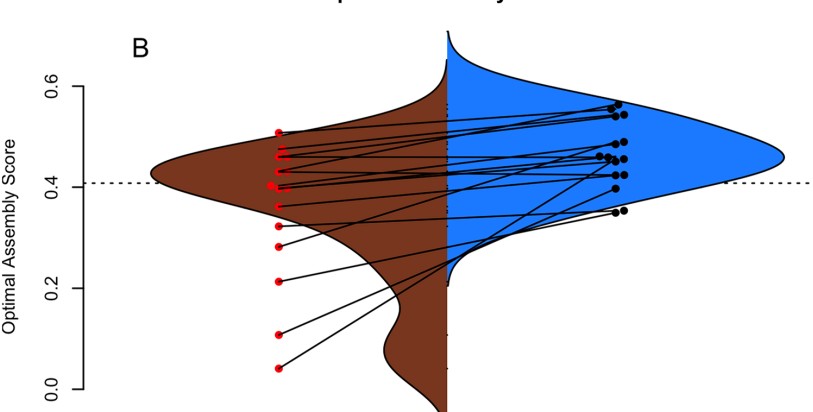

**Detonate Score**

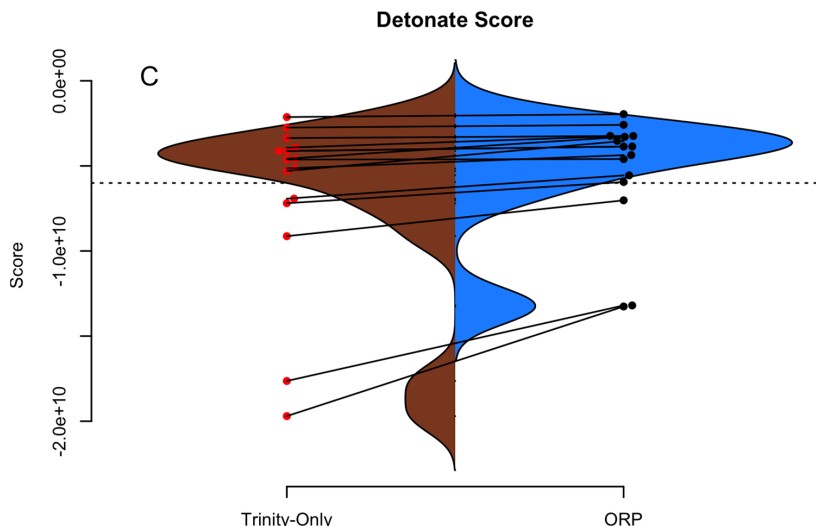

**Figure 2** **TransRate and Detonate generated statistics.** (A–C) Split violin plots depict the relationship between Trinity assemblies (brown color) and ORP produced assemblies (blue color). Red and black dots indicate the value of a given metric for each assembly. Lines connecting the red and black dots connect datasets assembled via the two methods.               

from the biological sample under study, when reads fail to map, that fraction of the biology is lost from all downstream analysis and inference. This study demonstrates that across a wide variety of taxa, assembling RNAseq reads with any single assembler alone may result in a decrease in mapping rate and in turn, the lost ability to draw conclusions from that fraction of the sample.

Figure 2B describes the distribution of `TransRate` assembly scores, which is a synthetic metric taking into account the quality of read mapping and coverage-based statistics. The `Trinity` assemblies had an average optimal score of 0.35 (sd = 0.14), while the ORP assembled datasets had an average score of 0.46 (sd = 0.07). This test is statistically significant (one-sided Wilcoxon rank sum test, $p$-value = $1.8E$-2). Optimal scores of the other assemblies are less than that of the ORP assembly, but in most cases, greater than that of the `Trinity` assembly. Figure 2C describes the distribution of `Detonate` scores. The `Trinity` assemblies had an average score of $-6.9E9$ (sd = $5.2E9$), while the ORP assembled datasets had an average score of $-5.3E9$ (sd = $3.5E9$). This test not is statistically significant, though in all cases, relative to all other assemblies, scores of the ORP assemblies are improved (become less negative), indicating that the ORP produced assemblies of higher quality.

In addition to reporting synthetic metrics related to assembly structure, `TransRate` reports individual metrics related to specific elements of assembly quality. One such metric estimates the rate of chimerism, a phenomenon which is known to be problematic in de novo assembly (*Ungaro et al., 2017*; *Singhal, 2013*). Rates of chimerism are relatively constant between all assemblers, ranging from 10% for the `Shannon` assembly, to 12% for the `SPAdes75` assembly. The chimerism rate for the ORP assemblies averaged 10.5% (± 4.7%). While the new method would ideally improve this metric by exclusively selecting non-chimeric transcripts, this does not seem to be the case, and may be related to the inherent shortcomings of short-read transcriptome assembly.

Of note, consistent with all short-read assemblers (*Ungaro et al., 2017*), the ORP assemblies may not accurately reflect the true isoform complexity. Specifically, because of the way that single representative transcripts are chosen from a cluster of related sequences, some transcriptional complexity may be lost. Consider the cluster containing contigs {AB, A, B} where AB is a false-chimera, selecting a single representative transcript with the best score could yield either A or B, thereby excluding an important transcript in the final output. I believe this type of transcript loss is not common, based on how contigs are scored (Table 1; Fig. 3; *Smith-Unna et al., 2016*), though strict demonstration of this is not possible, given the lack of high-quality reference genomes for the majority of the datasets. More generally, mapping rates, `Detonate` and `TransRate` score improvements suggest that this type of loss is not widespread.

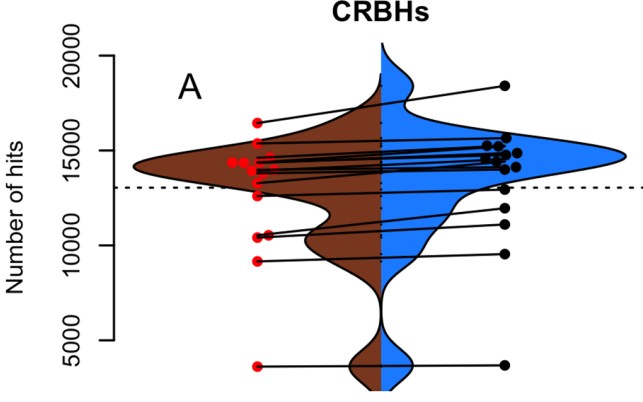

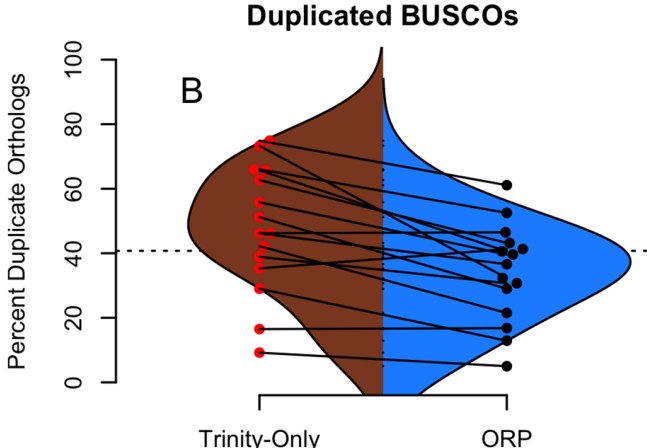

**Figure 3 Shmlast and BUSCO generated statistics.** (A and B) Split violin plots depict the relationship between Trinity assemblies (brown color) and ORP produced assemblies (blue color). Red and black dots indicate the value of a given metric for each assembly. Lines connecting the red and black dots connect datasets assembled via the two methods.

### Assembly content

The genic content of assemblies was measured using the software package `Shmlast`, which implements the conditional reciprocal blast test against the Swiss-prot database. Presented in Table 2 and in Fig. 3A, ORP assemblies recovered on average 13,364 (sd = 3,391) blast hits, while all other assemblies recovered fewer (minimum `Shannon`, mean = 10,299). In every case across all assemblers, the ORP assembler retained more reciprocal blast hits, though only the comparison between the ORP assembly and `Shannon` was significant (one-sided Wilcoxon rank sum test, $p = 4E\text{-}3$). Notably, in all cases, each assembler was both missing transcripts contained in other assemblies, and contributed unique transcripts to the final merged assembly (Table 2), highlighting the utility of using multiple assemblers.

Regarding BUSCO scores, `Trinity` assemblies contained on average 86% (sd = 21%) of the full-length orthologs as defined by the `BUSCO` developers, while the ORP assembled datasets contained on average 86% (sd = 13%) of the full length transcripts. Other

**Table 2 Describes the number of genes contained in the assemblies, with the row labeled concatenated representing the combined average (± standard deviation) number of genes contained in all assemblies of a given dataset.**

| Assembly | Genes | Delta | Unique |
|----------|-------|-------|--------|
| Concatenated | 14,674 ± 3,590 | | |
| SPAdes55 | | −1,739 ± 758 | 570 ± 266 |
| SPAdes75 | | −2,711 ± 2,047 | 301 ± 195 |
| Shannon | | −4,375 ± 3,508 | 302 ± 241 |
| Trinity | | −1,952 ± 803 | 520 ± 301 |

**Note:**
The other rows contain information about each assembly. The column labeled delta contains the average number (± standard deviation) of genes missing, relative to the concatenated number. The unique column contains the average number of genes (± standard deviation) unique to that assembly.

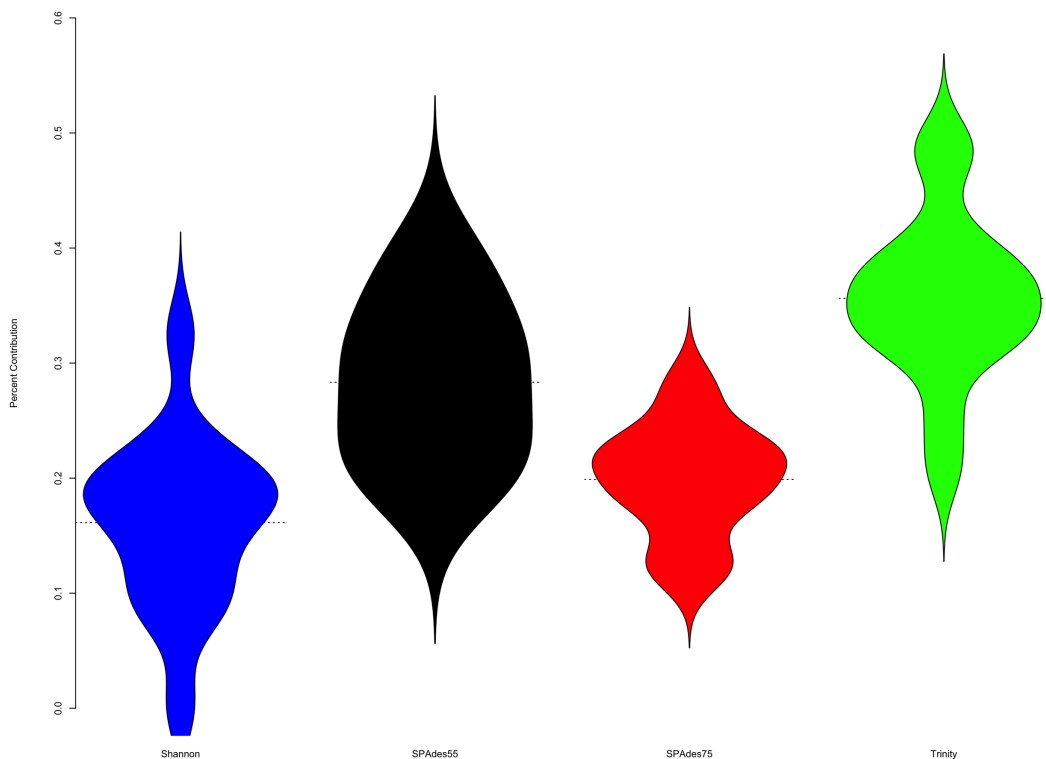

**Figure 4 Plot describes the percent contribution of each assembler to the final ORP assembly.** The proportion of the final transcripts contained in the merged assembly that are a product of each assembler is shown. Violin plots illustrate that Shannon contributes on average the fewest number of transcripts (<20% of transcripts) to the final merged assembly, while Trinity contributes on average the most. Small dashed lines on each side of the plot mark the median of the distribution.

assemblers contained fewer full-length orthologs. The `Trinity` and ORP assemblies were missing, on average 4.5% (sd = 8.7%) of orthologs. The `Trinity` assembled datasets contained 9.5% (sd = 17%) of fragmented transcripts while the ORP assemblies each contained on average 9.4% (sd = 9%) of fragmented orthologs. The other assemblers in all cases contained more fragmentation. The rate of transcript duplication, depicted in Fig. 3B is 47% (sd = 20%) for `Trinity` assemblies, and 34% (sd = 15%) for ORP

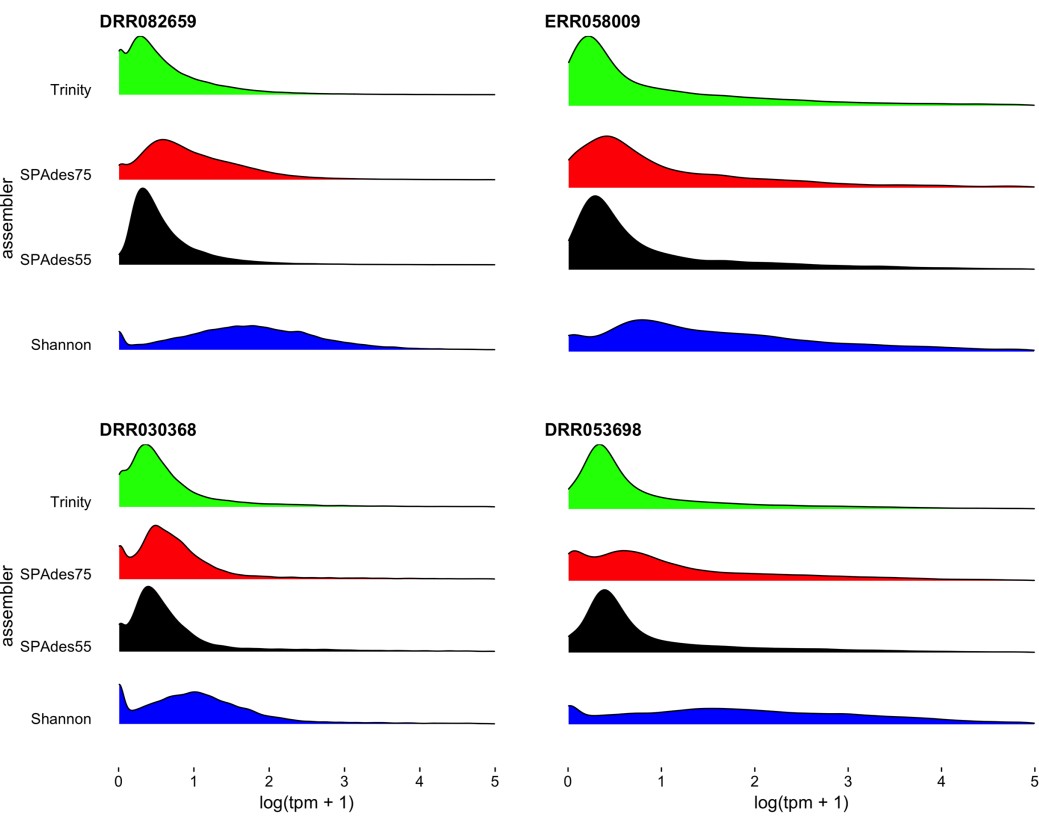

**Figure 5 Distribution of gene expression for each assembler.** Distribution of gene expression (log(TPM+1)), broken down by individual assembly, for four representative datasets are shown. As predicted, the use of a higher kmer value with the SPAdes assembler resulted in biasing reconstruction toward more highly expressed transcripts. Interestingly, Shannon uniquely exhibits a bias towards the reconstruction of high-expression transcripts (or away from low-abundance transcripts).

assemblies. This result is statistically significant (One sided Wilcoxon rank sum test, $p$-value = 0.02). Of note, all other assemblers produce less transcript duplication than does the ORP assembly, but none of these differences arise to the level of statistical significance.

While the majority of the BUSCO metrics were unchanged, the number of orthologs recovered in duplicate (>1 copy), was decreased when using the ORP. This difference is important, given that the relative frequency of transcript duplication may have important implications for downstream abundance estimation, with less duplication potentially resulting in more accurate estimation. Although gene expression quantitation software (*Patro et al., 2017*; *Bray et al., 2016*) probabilistically assigns reads to transcripts in an attempt at mitigating this issue, a primary solution related to decreasing artificial transcript duplication could offer significant advantages.

### Assembler contributions

To understand the relative contribution of each assembler to the final merged assembly produced by the ORP, I counted the number of transcripts in the final merged assembly

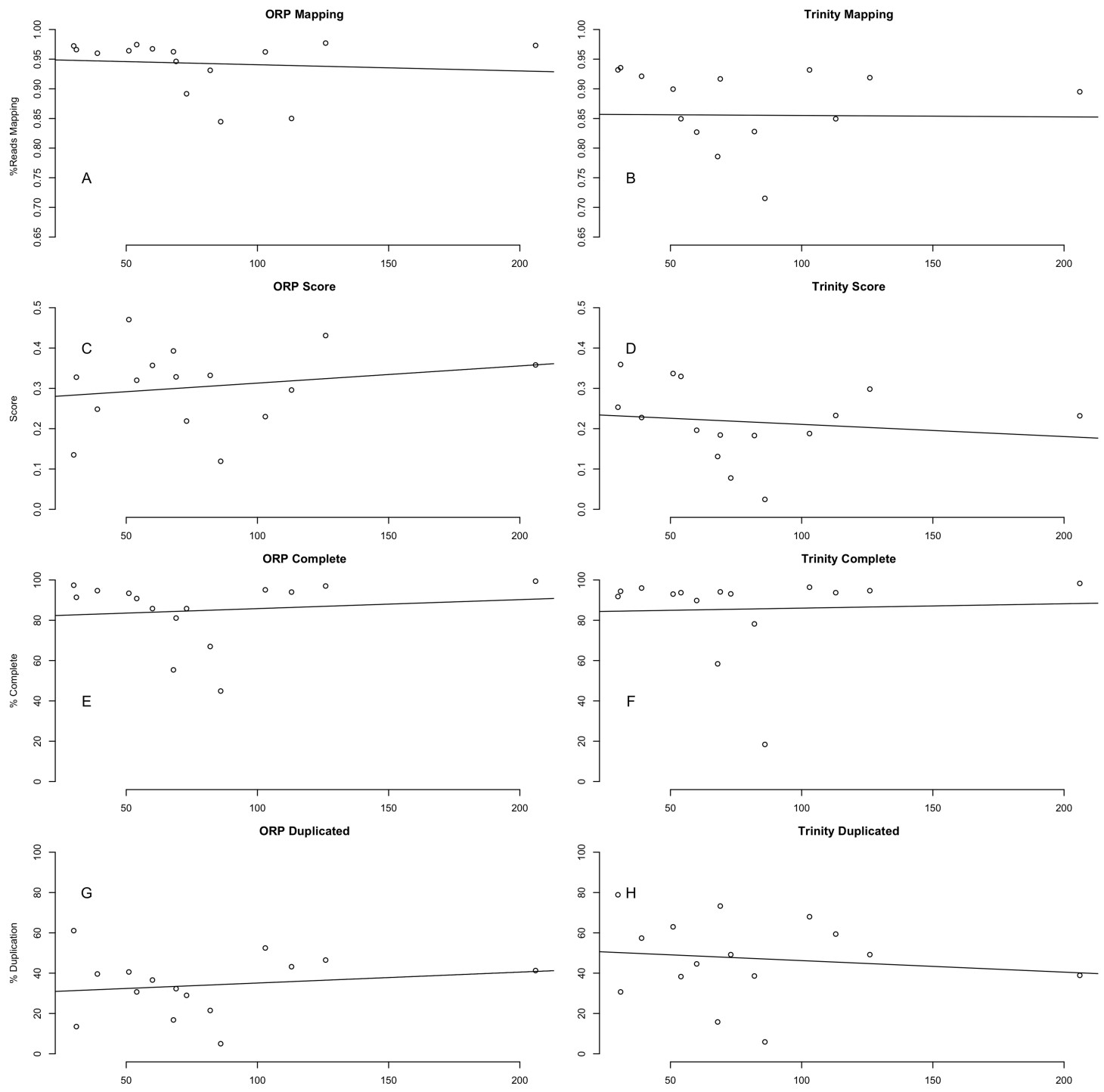

**Figure 6** **No relationship between metrics and dataset size.** The relationship between a subset of assembly metrics and the number of read pairs are shown and is not significant. (A) ORP mapping; (B) Trinity mapping; (C) ORP score; (D) Trinity score; (E) ORP complete; (F) Trinity complete; (G) ORP duplicated; (H) Trinity duplicated. In all cases the *x*-axis is millions of paired-end reads.

that originated from a given assembler (Fig. 4). On average, 36% of transcripts in the merged assembly were produced by the `Trinity` assembler. A total of 16% were produced by `Shannon`. `SPAdes` run with a kmer value of length = 55 produced 28% of transcripts, while `SPAdes` run with a kmer value of length = 75 produced 20% of transcripts.

To further understand the potential biases intrinsic to each assembler, I plotted the distribution of gene expression estimates for each merged assembly, broken down by the assembler of origin (Fig. 5, depicting four randomly selected representative assemblies). As is evident, most transcripts are lowly expressed, with `SPAdes` and `Trinity` both doing a sufficient job in reconstructing these transcripts. Of note, the `SPAdes` assemblies using kmer-length = 75 is biased, as expected, toward more highly expressed transcripts relative to kmer-length 55 assemblies. `Shannon` demonstrates a unique profile, consisting of, almost exclusively high-expression transcripts, showing a previously undescribed bias against low-abundance transcripts. These differences may reflect a set of assembler-specific heuristics which translate into differential recovery of distinct fractions of the transcript community. Figure 5 and Table 2 describe the outcomes of these processes in terms of transcript recovery. Taken together, these expression profiles suggest a mechanism by which the ORP outperforms single-assembler assemblies. While there is substantial overlap in transcript recovery, each assembler recovers unique transcripts (Table 2; Fig. 5) based on expression (and potentially other properties), which when merged together into a final assembly, increases the completeness.

## Quality is independent of read depth

This study included read datasets of a variety of sizes. Because of this, I was interested in understanding if the number of reads used in assembly was strongly related to the quality of the resultant assembly. Conclusively, this study demonstrates that between 30 million paired-end reads and 200 million paired-end reads, no strong patterns in quality are evident (Fig. 6). This finding is in line with previous work (*MacManes, 2015*), suggesting that assembly metrics plateau at between 20 and 40 M read pairs, with sequencing beyond this level resulting in minimal gain in performance.

### Funding
The author received no funding for this work.

### Competing Interests
The author declares that they have no competing interests.

### Author Contributions
- Matthew D. MacManes conceived and designed the experiments, performed the experiments, analyzed the data, contributed reagents/materials/analysis tools, prepared figures and/or tables, authored or reviewed drafts of the paper, approved the final draft.

## Data Availability
GitHub: https://github.com/macmanes-lab/Oyster_River_Protocol

Zenodo: DOI 10.5281/zenodo.1320141.

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
