# Peer review of "The Oyster River Protocol: a multi-assembler and kmer approach for de novo transcriptome assembly"

_PeerJ, doi:10.7717/peerj.5428_

## Round 0.1 · original submission · Major Revisions

I've elicited detailed criticisms from three world experts on transcriptome assembly and analysis; I don't think I could have gotten more competent critics if I had been doing this editing for any other journal. Please address all of the issues that they raise, in your revisions to the manuscript, and please answer all of their points in detail with your rebuttal letter. In particular, please note that these three different sets of comments have substantial overlap.

Some points that I noted, in my own reading of the article:

The entire protocol is designed to optimize for TRANSRATE. On one hand, you are correct to want to do this. On the other hand, the fact that you can get an improved transcriptome by the criterion that you are using to select possible cDNAs is not fully satisfactory, because it is circular. I think Brian Haas' suggestion here is an excellent remedy: continue using TRANSRATE, but cross-check your results with DETONATE.

There are a lot of small typographical errors, solecisms, awkward sentences, and statements that data are available 'here' (but with no URL provided). Kill them all! Burn them with fire! Yes, minor errors are intellectually trivial, but they have the cumulative effect of making the paper difficult to read and understand. They also run the risk of making broken trains of thought easy to overlook. Clear writing does not guarantee correct thinking, but it absolutely *does* make incorrect thinking easier to spot.

Validation on a well-defined, model organism gene set would be a great idea. Two different reviewers suggested mouse. That would be highly desirable.

In your table of the input RNA-set data sets that you tested, please order the data sets by alphabetical listing of the common phylogenetic class ("insect", "flatworm", etc.). This makes it easier for the reader to get a sense of what the overall composition is. Also, in subsequent figures/tables, please give a taxonomic name as well as an SRA accession number for the specfic data being shown.

Not necessary, but I would like this, and others might too: what is the origin of the term "Oyster River Protocol"? Can you provide an etymology in the Introduction? It is not scientifically necessary, but such touches help the reader remember the scientific content (weird, but I think true).

Good luck, and please do get this resubmitted as a seriously revised paper. There are a lot of criticisms, but the reason I could get three top reviewers for your work is that the field genuinely needs something like this. In a mature form, ORP will be a highly valuable resource.

·

Basic reporting

This is really interesting paper that I feel could yield good results however this manuscript feels rushed to me and I don’t think it is ready for publication yet.

Some of this is pedantic detail regarding the manuscript; the English needs to be be improved (e.g. “Troublesome, this biology is lost from all downstream analysis and inference”) as could the flow of the manuscript. There is considerable repetition (for example, at the start of the Discussion and the Conclusions) and p-values are routinely quoted to more significant figures than I believe (e.g. Section 3.2.1 p=0.0001322 for a comparison of the distributions with only 15 data-points per distribution). There are more serious shortcomings to the manuscript however that I think require considerably more attention.

I understand that the manuscript is a short presentation of a potentially useful tool but I do not feel that the authors give sufficient context in the introduction. In particular, there is no attention given to previous work on attempting to benchmark and reconcile the differences between potential de novo transcriptome assemblers, no attempt made to provide a list or or reference to the current suite of tools or any discussion of previous work highlighting how sensitive these tools are to a sensible and carefully chosen set of parameters (although the authors do correctly note that the apparent ease of use of these tools belies the complexity of the process).

On the plus side, the manuscript is well set out and the author has shared their code, the data they use is public and the assemblies will be made available, which is all good news.

Experimental design

The research question is well laid out, however the lack of summarisation of previous efforts to benchmark de novo transcriptome assembly makes it a little unclear how significant the problem they are trying to address is.

A good effort has been made to assess the performance of their new protocol in the context of other stand-alone assemblers, but I am concerned about several aspects of the process:

1) The main standalone tool they compare the Oyster River Protocol (ORP) to is Trinity, which itself forms one of the three assemblers that comprise the ORP. As such, it is perhaps not surprising that the ORP might outperform it.

2) The author doesn’t really give any justification for why he chose these three tools for the ORP as opposed to any other of the myriad of different tools. Did he choose them because they were popular, or known to give good results, or because they have very different underlying algorithms and so might be sampling different regions of search-space?

3) The data preparation for the stand alone Trinity runs is not the same as for the ORP assemblies. Specifically, the data for the stand-alone Trinity runs doesn’t include the error correction step that is included in the ORP from a previously published tool (RCorrector). As a result, it is unclear whether the improvement observed for the ORP over Trinity is a result of combining the assemblies or the error correction. In particular the % read -mapping Transrate score - which is where the ORP sees the most dramatic improvement over Trinity - may well be be strongly affected by the error correction step.

4) The manuscript doesn’t give very much detail on how the BUSCO transcripts were assessed. For example, the manuscript often includes statements such as “Trinity assemblies contained on average 86% (sd = 21%) of the full-length orthologs, 154 while the ORP assembled datasets contained on average 85% (sd = 16%) of the full length transcripts.” without carefully describing what this actually means. Are these isoform matching with 100% identity over 100% of both the ISOFORM and BUSCO transcript length? I highly doubt that the matches are this perfect (from my own experience with de novo transcript assembly). We really need considerable more detail here on how these things are compared and considered matching or not.

5) The inclusion of multiple kmer lengths is interesting, but no real effort is made to determine whether the differences between the individual assemblies is due to the different kmer lengths or the different algorithms of the tools. For example, how does an ORP assembly compare with three Trinity assemblies, with kmers of 25,55 & 75, combined in the same fashion?

Overall, however, the manuscript is lacking the depth and detail I think it needs to really fill the knowledge gap and without this details reproducing these results would be difficult despite the easy and availability of the software install and the dataset.

Validity of the findings

In its current form, the manuscript has not convinced me that the ORP represents any significant improvement over Trinity, the existing stand-alone de novo assembly tool tested. As such, I think that the conclusion the the ORP represents an improvement over existing tools is not yet sound.

Additional comments

This work details a new multi-assembler approach to improving de novo transcriptome assembly. This is a really good idea that is worth exploring and I really want to see a strong detailed exploration of the concept and how much we might gain from combining de novo transcriptome tools, but I’m afraid that these have not been adequately illuminated here. A little more time, exploration of the tools and more careful analysis and I think it would make a good paper. I'd be happy to review future revisions of the paper and would like to see ORP bradened to include more assembly tools.

The instruction for installing and running the software (http://oyster-river-protocol.readthedocs.io/en/latest/aws_setup.html) work and I could set up the tool on my local box and verify that it runs. Unfortunately I don't have access to sudo on my clusters large memory machines, and we have no way of charging AWS tie to grants, so I couldn't test it on live data. Perhaps, given the complex web of dependencies involved, it might be an idea to distribute this as a conda package or a docker instance?

Reviewer 2 ·

Basic reporting

The manuscript is well written and well organized.

The transcriptome assemblies resulting from each of the assemblies are made available as supplementary data, but there are many other supplementary data files and figures that would be useful to include that were not available, such as the TransRate reports for each of the assemblies, and ideally a summary data table that includes all TransRate scores and score components (ie. the individual metrics computed by TransRate that are leveraged in computing the final overall assembly score). Busco scores for all assemblies should be included as well.

Although multiple transcriptome assemblers were leveraged, the figures focused almost entirely on comparisons between the single assembler Trinity and the author's results from ORP. It would be useful to include comparisons to the other assemblers as well in a similar format, and provided as supplementary materials if not deemed worthy as main figures.

Experimental design

The ultimate goal of ORP is to generate a final assembly that contains the best assembled transcripts among all leveraged assemblers and to generate a final mixed assembly that is of higher quality than any of the individual input assemblers. Specifically, the ORP method involves running multiple assemblers, scoring each transcript, and clustering together transcripts that likely correspond to the same isoform of each gene. Based on TransRate contig scores, the single highest scoring transcript within each group is selected to represent that group in the final ORP reported assembly.

Comparisons among the final assemblies involve the TransRate final assembly scores, percent of reads represented by each assembly, and core ortholog representation as per Busco. These are all useful metrics, but it would be of interest to include additional metrics that reflect rates of chimerism and correct representation of alternatively spliced transcripts. In general, accuracy evaluations should rely on two metrics: sensitivity and specificity. Most metrics leveraged by the author nicely reflect sensitivity, but there is little devoted to specificity. Targeting specificity is difficult to do with data sets lacking reference data sets, but there are many well-studied model organisms that can be leveraged for this analysis, and among the data sets leveraged by the author, mouse is included - which has a high quality reference genome and reference annotation set where evaluation of correct alternatively spliced isoforms could be explored. Simulated data sets are also very useful for evaluating performance metrics, given that the reference transcripts (and splicing isoforms) are known entities - and in resulting assemblies, incorrectly assembled transcripts, chimeras, or missing splicing isoforms can be readily identified.

It would benefit the manuscript to include an orthogonal measure of assembly quality such as provided by the DETONATE software. Ideally, this would show that ORP compiled assemblies have improved DETONATE scores as compared to the assemblies provided as input to ORP.

While ORP attempts to stringently cluster transcripts into isoform-level groupings as a way of minimizing the loss of relevant splicing isoforms, there's no evidence shown of the impact of this method on splice isoform representation in the final ORP assembly. This deserves much more attention in the manuscript. Also, in the case of isoforms that are fused as chimeras, selecting a single best representation from a cluster containing chimeras and non-chimeras would expect to contribute to loss of important transcripts. For example, given a cluster {AB, A, B} where AB is a chimera, selecting a single representative transcript with the best score would presumably yield A or B and hence exclude an important transcript in the final output. More attention to resolving clusters of transcripts containing chimeras is warranted.

Since ORP is centered around combining results from multiple assemblers, it begs the question of what assemblers would be generally best suited to use with ORP. The author leverages Trinity, Shannon, and Spades. Trinity and Shannon have been previously demonstrated to be highly effective transcriptome assemblers, but to my knowledge, Spades has seen comparably little use for transcriptome assembly; instead Spades has been more targeted towards genome assembly. There does appear to be an rnaSPAdes method included with Spades that's more specifically targeted to transcriptome assembly, but I could find little information about it being used in practice nor how effective it is for resolving spliced isoforms, which is one of the primary goals of transcriptome assembly and one that most differentiates it from other assembly challenges. One of the goals of ORP is to leverage multiple kmer lengths as part of the assembly, and Spades is run twice using different kmer values as one way to accomplish this. Using multiple kmer values to generate an assembly, leveraging the benefits of small and large kmers, is a feature of existing transcriptome assemblers Oases and IDBA-tran, neither of which were used by ORP nor cited. It isn't clear as to why the author would rely so heavily on Spades when other highly effective transcriptome assemblers are readily available and already implement ideas that are central to ORP's goals. ORP would ideally be shown to benefit from using the very best assemblers as well as to be resilient towards including results from assemblers that are sub par.

Validity of the findings

The primary finding is that the author's ORP yields assemblies that are generally higher in quality than the Trinity assembler alone, primarily leveraging TransRate and BUSCO for assembly quality assessment. In terms of the bulk statistics, this is a valid conclusion, but it does not specifically address reducing chimeric contigs and ensuring representation of alternatively spliced isoforms, both very important aspects of transcriptome assembly and also often negligible with respect to the bulk statistics focused upon here.

Overall, the results section is overly directed towards comparisons to Trinity when there were several other assembly methods being used by ORP. For each assembly quality metric, it would be useful to see how ORP compares to each of the individual assemblies, including Trinity. If there exist cases where any individual assembly used as input to ORP ends up outperforming ORP, it would call into question the value of applying ORP as a general protocol.

Additional comments

I commend the author on the development of ORP. Generating high quality transcriptome assemblies is an important challenge, and ORP encapsulates many of the ideas that are highly relevant towards achieving this goal. I have several constructive recommendations towards ensuring that ORP demonstrates that it achieves its aims:

1. ORP should ideally outperform each of the individual methods used as input. Performance should include reduced chimerism and retention of alternatively spliced isoforms that are well supported by the rna-seq data.

2. ORP should be resilient to including 'bad' assemblies along with 'good' assemblies as input.

3. ORP should demonstrate improved assembly quality across metrics that address both sensitivity and specificity (eg. leveraging DETONATE along with TransRate, and leveraging high-quality model organism reference data sets).

Additional comments:

As a protocol, ORP should apply equally well to de novo transcriptome assemblies as to genome-guided assemblies. Using reference model organism data sets, one could explore applying ORP to transcript sets resulting from cufflinks, stringtie, or other genome-guided methods such that ORP will yield the best quality transcript set.

Most data sets in the manuscript appear to lightly benefit from ORP, whereas there are a few assemblies that greatly benefit from ORP, especially in the context of percent reads mapping. It would be useful to comment on this and whether the suboptimal Trinity assemblies in these cases were reflective of some aspect of the data, and whether the alternative assemblers provided much better assemblies than Trinity in these cases. This would be much easier to assess if the results from all methods were easily accessible in tables and figures in supplementary materials.

The composition of the final ORP assemblies appears to mostly derive from Spades assembled transcripts. I imagine this could happen for several reasons. For example, (a) there could be clusters of transcripts where Spades yields the best representative transcript with the highest score, (b) there are clusters where there are tied scores and Spades is chosen preferentially, (c) Spades generates many transcripts that are unique to Spades and these are automatically propagated to ORP. It would be useful to know how the composition of the final ORP assembly reflects which transcripts were selected as best within clusters vs. alternative justifications for their inclusion in the final assembly.

It would be useful to include in main figures or supplementary materials examples of genes that are better represented by the output of ORP as compared to any one individual assembler, having captured proper splice variants and/or resolved chimeras.

·

Basic reporting

See below.

Experimental design

See below.

Validity of the findings

See below.

Additional comments

# Review of Oyster River Protocol

In this paper, Dr. MacManes describes a new computational protocol for de novo transcriptome protocol. Briefly, the protocol first runs three different assemblers (Trinity, Shannon, and SPAdes) with multiple parameters, combines the resulting assemblies and extracts ortholog groups using a custom modification of OrthoFinder called OrthoFuse, and finishes using a custom version of Transrate to select the highest contig score from each ortholog group. Dr. MacManes demonstrates that there are very few downsides to using this protocol over e.g. a standard Trinity protocol: many metrics are either the same or improved.

This is an important paper for a wide range of biologists, because (as is correctly noted in the paper) de novo transcriptome assembly is very widely used in research into the biology of many (most!) different eukaryotic organisms. The completeness of reference transcriptomes is important for all the downstream analyses, including annotation, recovery of homologs, and inference of differentially expressed genes.

Along these lines, there are a number of conclusions in the paper that are quite interesting and important:
* mapping rates and transrate assembly scores are significantly improved in the ORP assemblies.
* Busco completeness scores are generally the same or improved in the ORP assemblies.
* the different assemblers appear to be better at recovering transcripts from different abundance fractions of the reads (which I find really interesting and unexpected, at least in the degree to which it seems to occur). This observation will likely lead to methodological advances in standalone assemblers, which is excellent.

The work in general seems to be of high quality and many of the conclusions are solidly grounded in the results and discussion.

However, there are a number of puzzling choices and omissions in the paper as it stands. I detail these below. There are also many minor typos and grammatical errors that I will communicate directly to the author.

Questions and major comments:

* line 52 - I would imagine that other kinds of biases might result from the approach chosen in the paper; any thoughts as to downsides of the ORP approach here? (See comment on evaluating against a quality ref transcriptome, below.) e.g. in genome assemblies, maximizing N50 leads to misassemblies; might that happen here?

* unweighted assembly content is not systematically evaluated in this paper - most of the evaluations such as mapping rate and transrate score are biased by underlying abundance distribution of the transcripts, i.e. more abundant transcripts will produce more reads. I would suggest doing some kind of unweighted assembly comparison (one with higher resolution than BUSCO; next point) to see if more overall content is produced. This could be what Figure 4 intends to show, at least partly, but see below.

* it seems that no evaluation is done against a quality reference transcriptome, e.g. mouse RNAseq -> mouse. This is a major omission; the major reference-based approach used, BUSCO, only looks at a small subset of the transcripts, and does not evaluate transcript content systematically. It would be valuable to use a reference-based measure of transcriptome quality here, such as the Trinity paper used.

* The choice of taxonomic descriptions in table 1 is bewildering - I don't see any reason (biological or other) for the descriptions chosen. Maybe stick with 'animal' and 'plant'?

* More, section 4.2 incompletely addresses an interesting question - there are many features of genomes/transcriptomes that are more likely to challenge transcriptome assembly than taxonomic membership, e.g. tissue sampled, tissue complexity in terms of number of cell types, presence of RNA editing, recent genome duplications, polymorphism rate, etc. It's probably beyond the scope of this paper to explore these issues, but I'm not sure how it merits the incomplete discussion it's given in the current paper. Perhaps a single sentence "ORP performed equally well across all of the organisms chosen at random" would be better, or something like that.

* The paper states that no value is seen in including reads beyond 30m, which is intuitive and matches our experience. However, most transcriptome assembly projects sequence multiple tissues and then combine them in a de novo assembly. Is the recommendation for the ORP to subsample each tissue data set to 30m and then combine them? Or what? Some comment (along with a specific reference to a paper with an overall suggested workflow for making use of the assembly) would be welcome.

* Along those lines, the extra computational requirements of the ORP seem likely to be significant when combining multiple tissue RNAseq data sets. This is one of the reasons that Trinity recommends using in silico normalization, which was explicitly *not* used in this paper. The discussion should be expanded to include this.

* I'm puzzled by the division between Results and Discussion in this paper. I'd suggest either combining them or having the primary results be separate from the discussion of what the results mean; typically the division I use is "here are the facts I see" (for results), and "here is what I think they mean".

* I had trouble understanding the section starting at line 188, and figure 4, at least at first. If I understand correctly, this paragraph is discussing both swissprot match percentages AND transcript "origin" percentages. The latter can be studied because orthofuse doesn't actually merge transcripts, it picks a "best" transcript from each orthologous grouping - is that right? The former (swissprot analysis) is of objective interest because it speaks to what genes are being recovered by which assembler. Or... maybe that's not what the paragraph is saying. It would be nice if this could be clearly split into discussions of homology/orthology/recover of known genes, and recovery of transcripts (known or unknown). As it is I think it's muddled in its message. Figure 4 seems clear enough but I'm not sure if it's talking about transcripts (my guess) or swissprot matches. (In retrospect, it seems like the beginning of section 3.2.3 is talking about transcript origin wrt the final assembly, while the bottom of that section is talking about swissprot).

Minor comments:

* surely there is a review citation of some sort for the statement that "transcriptome sequencing has been influential"? p2 line 24
* Throughout the paper, clickable links in the PDF are used (The code ... is available _here_.) these should be replaced with URLs, perhaps in footnotes or references.

---

## Round 0.2 · Minor Revisions

It is very encouraging to see that two of the three reviewers both agreed that the paper is qualitatively improved and that their criticisms have been addressed. Please do revise the paper further to address their remaining criticisms. In the event that these final criticisms are dealt with, I am confident that your paper will be accepted.

I apologize for the delay in getting this verdict to you. The paper was resubmitted immediately before Christmas vacation, a time period when none of my reviewers were in a position to work on the re-review. There were other factors over which I had no control that discouraged one of the three reviewers from a follow-up at all. Of the two reviewers who did respond, neither was in a position to do so quickly. So, it is not the case that your need for a reviewed paper has been ignored for months (though, if I were in your shoes, I would certainly be feeling that way by now...)

·

Basic reporting

The overall quality of the language and the structure of the paper is much improved and it now reads much better.

Experimental design

I am happy to see that the paper has been considerably improved based on the three reviewers comments. The paper now demonstrates more conclusively not that there is a small, but clear, improvement in the quality measures of assemblies constructed using the ORP over stand-alone Trinity assemblies.

I'm still not entirely convinced that the relatively small improvement is worth the extra compute time and resources, but I am convinced its real and that some researchers will find ORP useful. The tool has been made easy to use and the difference between a 16h run of Trinity and a few days with ORP is really not a huge difference in the scheme of things.

Validity of the findings

I am happy with the paper in this category.

I would suggest that the authors include a statement reminding the reader that the one-sided Wilcoxon rank sum test is a non-parametric statistic and, as such, makes no assumptions about the nature of the distributions its testing. I think this is worth stating because its far from clear that the underlying distribution of things like the Transrate Scores are and that using a parametric test (such as a t-test) would not be justified.

Additional comments

The author has adequately addressed all the previous points I raised. I do still have a few minor (mostly cosmetic) comments that I'd like addressed further before publication. I'd recommend accepting the manuscript if these comments are addressed.

1) The author has included a justification for why he chose these three tools for the
ORP as opposed to any other of the myriad of different tools. I'd like to see a reference to the description of the characteristics of these tools in the introduction put in here so a reader who did not read the introduction knows where to look for this summary information!

2) All Figures: I don't like the axes used in many of the figures - axes should span the entire range of the data and have a defined intercept.

3) Figure 1: no axis labels - I realise the axes are not really meaningful for MDS plots but the labels should at least state what is being plotted - MDS parameter1 vs MDS parameter 2.

4) Figure 2 & 3: no axis labels (see above) - also might be better shown as a single tall figure with one column and two/three rows with the panels one above another and a shared x axis.

5) Figure 4: The captions doesn't really walk me through the figure so I'd like that expanded. Also, there are one very tiny lack points plotted on top of and underneath the violin plots - are these the data themselves, or some specific added dashed lines? What do these points mean? If they are intentional, they should be descried in the caption and made more obvious (thicker lines, large markers). If not, they should be removed.

6) Figure 5: I really don't like Joy plots (yes, I know they are the rage and people think they look cool, but...). Different panels of a figure in a scientific should not overlap - they should have their own labelled axis with a scale on. Only for a situation where the figure with separate panels would absolutely not fit on a page could I see an argument for Joy plots in a serious paper over just multi-panel plots - and we're not so tight on space here. These plots need re-drawing and multi-panel distribution plots.

7) The conclusion section is completely lacking in quantitative detail and repeats much of what has been said earlier in the paper. I'd suggest either removing it completely or, if you want to keep it (or the journal require it), then be precise and quantitative in it. Where does ORP do better and y how much does it do better.

Reviewer 2 ·

Basic reporting

In the revised manuscript, the author provides access to additional assembly metrics and data sets.

Experimental design

In the revised manuscript, the orthogonal method 'DETONATE' was leveraged to assess assembly quality, and results appear to be consistent with the TRANSRATE assessments.

Validity of the findings

There is good evidence for the ORP protocol to produce assemblies that are of higher quality than running Trinity alone.

Figure 1 is confusing and I question whether it is valid, as one would expect assemblies to cluster together according to organism type rather than by assembly method, and perhaps within each organism type, there might be some relationship among the methods with some being more similar to others. Instead, the distance metric appears to be showing a large batch effect where the primary difference is the program being used. Quite surprising...

Additional comments

I commend the author for the revisions, as the manuscript is greatly improved from the earlier submission.

In Table 2, the ORP method should be included as one of the comparators. If there are important genes that are found in the concatenated assembly but missing from the ORP assembly, it would be useful to explore why and comment accordingly.

Please check the revised version carefully for additional grammar and spelling errors.

---

## Round 0.3 · accepted · Accept

The previous Academic Editor handling your submission has stepped down from the Board, so I was asked to make the decision on this revision. I am happy to Accept this submission.

Richard Emes

#